# Chirally locked and dynamic bis-perylene diimide macrocycles with multiple sources of chirality

Denis Hartmann [1], Samuel E. Penty [2], Robert Pal[2] & Timothy A. Barendt [1] ✉

Chiral organic materials show great promise in optoelectronics, sensing and catalysis. Among those, macrocycles are of great interest due to their preorganisation and potential amplification of chiroptical properties. Understanding the effects of different sources of chirality on the resulting chiroptical properties of these molecules is key to unlocking tailored chiral materials. To this end, we have synthesised a family of bis-perylene diimide-based macrocycles containing multiple sources of chirality, specifically point chirality in the linker, helical chirality in the perylene diimide and supramolecular chirality in the macrocyclic dimer. We found a dominant effect from the helical chirality of the perylene diimide on the chiroptical properties, including the induction of chirality in an achiral guest molecule, which opens up new possibilities for hybrid chiroptical materials.

Chiral organic materials are under heavy investigation due to their use in chiral recognition and sensing[1], chiral optoelectronics[2], catalysis[3], and spintronics[4]. Among such materials, supramolecular assemblies are particularly exciting, as they show strong chiroptical properties[5] arising from amplification of chirality[6], as well as additional functionality, such as stimuli-responsiveness[7], including via guest binding[8,9], that can be incorporated into such systems.

A class of organic chromophores with burgeoning applications as (chiral) optoelectronic materials are perylene diimides (PDIs), which are valued for their stability, high quantum yields, ease of functionalisation and wealth of supramolecular chemistry[10,11]. Modification of the PDI core on the bay position provides a convenient handle for tuning of both photophysical properties[12] and self-assembly[13]. This also leads to helical core-twisting of the molecule, thereby introducing intrinsic helical chirality and the potential for chiroptical properties[14]. While this twisting generally weakens $\pi$-stacking interactions between PDIs[13], it unlocks additional functionality such as chiral guest binding[15,16] and circularly polarised luminescence (CPL)[17,18]. Their assembly into supramolecular structures may enable amplification of chiroptical properties via excitonic coupling[19], with macrocycles containing two PDI units being of particular interest as they allow the discrete study of supramolecular interactions and additionally enable guest binding in the macrocyclic cavity[20]. This guest binding can also allow for chiral imprinting with the aim of transferring chirality from one species to another, and has been realised in a PDI-based system with 1:2 host:guest stoichiometry[21]. It is important to understand the structure-property relationships that arise from the interaction between chiral twisted polycyclic aromatic hydrocarbons (PAHs) and their achiral planar counterparts to optimise systems

for chiroptical applications. Such mixed chiral/achiral chiroptical materials have already been developed, for example, via the incorporation of a chiral helicene within an achiral polymer matrix[22] or an achiral emitter interfaced with a chiral polymer[23].

The incorporation of core-twisted (i.e., *M* or *P*) PDIs into bis-PDI macrocycles allows for both homochiral (*MM*/*PP*) and heterochiral (*MP*) stereoisomers of the two interfacing dye units. If the PDI monomers are not chirally locked, these can interconvert through a somersault[17,24] through the macrocycle cavity. For persistent chiroptical properties, this process is undesirable. Therefore, to suppress this interconversion on bay-connected macrocycles, sterically-demanding imide substituents have been installed[18,25], whereas for imide-linked macrocycles, this has only been achieved through locking of the PDI's intrinsic helical chirality. Here, bulky tetraaryl bay-substituted PDIs[26] and macrocyclic strapping across the PDI faces have provided the required stability[16,21,27].

In this work, we aimed to prepare the first configurationally stable macrocycles composed of two chiral, core disubstituted PDIs connected via the imide positions and without additional covalent strapping over the PDI units, as has previously been employed in bis-PDI macrocycles[21,28]. This is achieved through 1,7-substitution of the bay-positions, which causes twisting, for chirality[14], and elongation of the aryl arms from phenyl to terphenyl, which prevents PDI rotation and thus affords configurational stability (Fig. 1).

The chiral elements in the prepared macrocycles include helical chirality in the PDI (arising from the core-twist, *M*/*P*) and supramolecular chirality of the PDI $\pi$–$\pi$ dimer (arising from their rotational displacement, $M_s$/$P_s$). We also included point chirality in the linker (via L-valinol-derived

[1]University of Birmingham, School of Chemistry, Edgbaston Campus, Birmingham, B15 2TT, UK. [2]University of Durham, Department of Chemistry, South Road, Durham, DH1 3LE, UK. ✉e-mail: t.a.barendt@bham.ac.uk

imide groups) to enable the isolation of macrocycle stereoisomers without chiral HPLC, as well as probing the effects of this additional source of chirality on the system. Previous imide-connected macrocycles only contained two of these elements, i.e., they did not contain helical PDI chirality[29], or supramolecular chirality[16].

We investigated the chiroptical properties of this family of macrocycles and found a dominant contribution from intrinsic helical PDI chirality to chiroptical properties. The macrocycles were also found to act as hosts for planar PAHs. The homochiral macrocycles were able to induce circular dichroism (CD) in the achiral planar guest, and showed no reduction in $g_{lum}$ upon guest binding, whereas the heterochiral macrocycle showed a loss of supramolecular chirality in both CD and CPL upon binding, emphasising

the importance of intrinsic PDI chirality for the development of chiroptical materials.

## Results and discussion
### Synthesis and characterisation

To prepare the desired macrocycles, dibromo-perylene dianhydride (as a mixture of 1,7 and 1,6 regioisomers)[30] was modified with L-valinol at the imide positions in DMA:dioxane, yielding the desired dibromo L-valinol-PDI 1. We attempted to separate the desired 1,7-regioisomer from the minor 1,6-regioisomer at this stage, but were unable to due to the relatively poor solubility of the compound. Therefore, the free alcohols were protected with *tert*-butyl silyl (TBS) groups, which drastically increased the solubility and allowed for the isolation of the 1,7-isomer 2 by preparative high-performance liquid chromatography (HPLC, Supplementary Fig. [Supp. Fig.] 1-1). Notably, the $^1$H NMR spectrum of the regioisomerically pure TBS-protected PDI 2 (as well as the 1,7 bay-arylated derivatives 3 and 6) showed doubling of the PDI core proton peaks, presumably due to *syn-* and *anti*-rotamers arising from restricted rotation about the imide N–C bond[31,32].

Suzuki-Miyaura cross-coupling of the dibromo-PDI with either phenyl- or terphenyl-boronic acid allowed for the installation of the desired bay aryl-substituents in quantitative yield. The TBS groups were then removed using HCl in Et$_2$O, yielding the bay-substituted diol-PDIs. These diols showed broad room-temperature $^1$H NMR spectra, which sharpened upon heating (Supp. Figs. 8-1 and 8-2), indicating an increased tendency for aggregation upon removal of the sterically bulky TBS groups.

The target macrocycles 5 and 8 were prepared by the slow addition of a solution of malonyl chloride via syringe pump into a solution of the appropriate diol and pyridine in CH$_2$Cl$_2$. The phenyl-modified macrocycle 5 was isolated following purification by preparative thin-layer chromatography (TLC). The $^1$H NMR spectrum for this macrocycle showed multiple sets of signals, most clearly visible for proton H$_a$ (Fig. 2c), indicating a

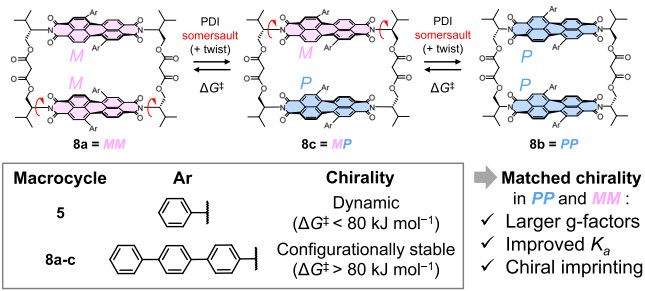

**Fig. 1 | Aryl-substituted, L-valinol-derived, imide-linked bis-PDI macrocycles were prepared in this work.** Macrocycle 5 with a short phenyl bay-substituent rapidly interconverts between stereoisomers via a somersault and twist mechanism (see Supp. Fig. 6-1 for more detail), whereas macrocycles 8a–c with extended terphenyl arms have stable chiral configurations, *MM*, *PP* and *MP*, respectively. Matched chirality between the two PDI units in the macrocycle (*MM*/*PP*) yielded larger g-factors, improved guest binding and the imprinting of chirality onto achiral guests.

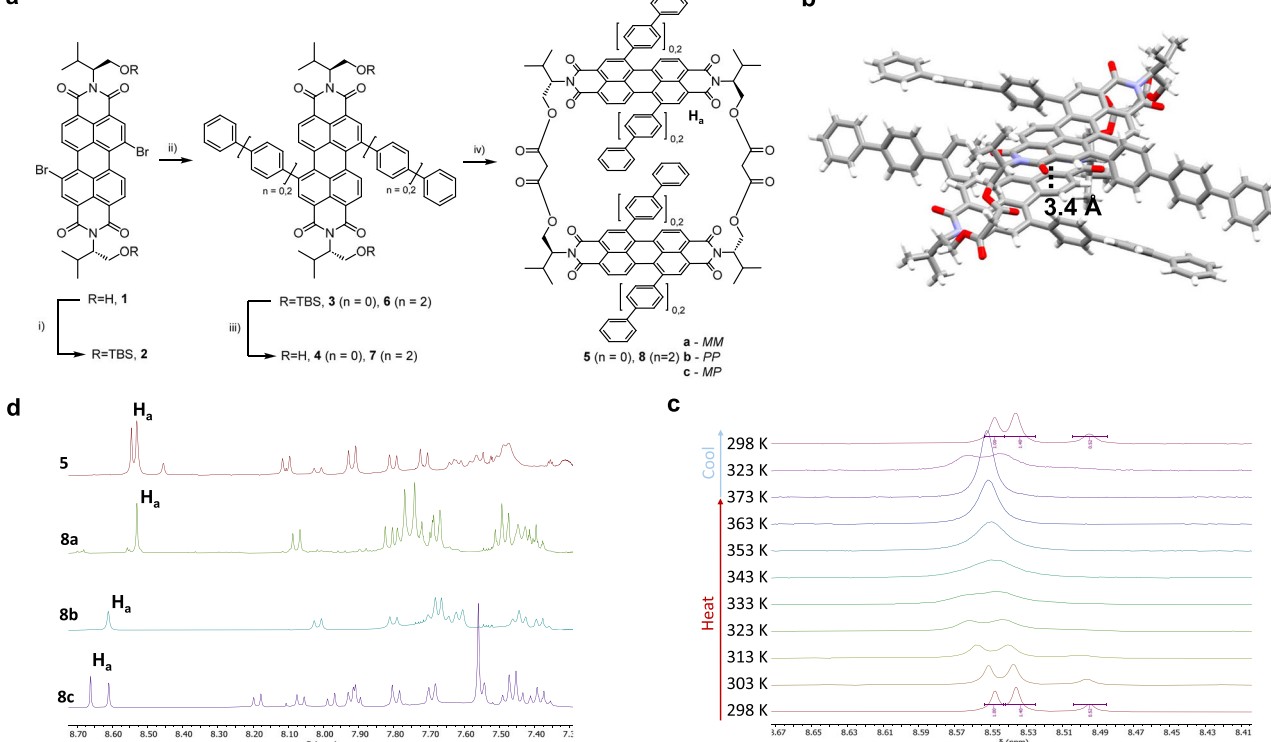

**Fig. 2 | Synthesis and characterisation of a family of core-twisted, aryl-substituted, L-val–bis-PDI macrocycles. a** Synthetic scheme. (i) TBSCl, imidazole, DMF. (ii) Ar-B(OH)$_2$, Pd(PPh$_3$)$_2$Cl$_2$, PhMe:EtOH:H$_2$O, K$_2$CO$_3$, 70 °C. (iii) HCl, Et$_2$O. (iv) Malonyl chloride, pyridine, CH$_2$Cl$_2$. **b** Single-crystal X-ray diffraction structure of *MP*-isomer 8c (solvent omitted for clarity). **c** $^1$H NMR spectra of the aromatic regions of macrocycles 5 and 8a–c in CDCl$_3$. **d**) Variable temperature $^1$H NMR spectra of macrocycle 5 proton H$_a$ in TCE-$d_2$.

mixture of $MM$, $PP$ and $MP$ chiral conformers, which all share a diastereomeric relationship due to the (point) chiral L-valinol-based linker. These diastereomers are not configurationally stable since, upon heating, the $^1H$ NMR peaks coalesced into a single set of signals (Fig. 2d, Supp. Fig. 8-3, $\Delta G^{\ddagger} = 78$ kJ mol$^{-1}$). The same ratio of $^1H$ NMR peaks was observed upon cooling, demonstrating that these diastereomers are in equilibrium. The mechanism of stereoisomer interconversion consists of two steps: a core-twist of a PDI unit as well as an intramolecular somersault through the cavity (Supp. Fig. 6-1)[18,25]. We calculated the barrier for core-twist interconversion of a Ph-substituted PDI monomer by DFT (Supp. Fig. 6-4 and Supp. Data 3–5), yielding a barrier of 34 kJ mol$^{-1}$, in line with other reported core-twist barriers for 1,7-bay-substituted PDIs[14,15]. Importantly, this barrier is lower than that measured by $^1H$ NMR spectroscopy ($\Delta G^{\ddagger} = 78$ kJ mol$^{-1}$), which confirms that inhibiting the PDI's intramolecular somersault is key to preventing bis-PDI macrocycle racemisation.

Therefore, by extending the length of the bay substituents, terphenyl-modified macrocycle **8** provides a strategy for realising chiral and configurationally stable PDI dimers. Three macrocyclic products were isolated by preparative TLC (Supp. Fig. 1-2), which correspond to the three $MM$, $PP$ and $MP$ diastereomers of the PDI dimer, macrocycles **8a**, **8b**, and **8c**, respectively (vide infra). We confirmed the configurational stability of **8a–c** by chiral HPLC (Supp. Fig. 3-1). While macrocycle **5** gave a single peak on the chromatogram, evidencing rapid interconversion of diastereomers, **8a–c** gave individual peaks with distinct retention times. Furthermore, $^1H$ NMR spectroscopy revealed that the ratio of $H_a$ proton peaks in the spectrum of **8c** is unchanged upon heating to 100 °C (Supp. Fig. 8-5), meaning that the diastereomers of the terphenyl-functionalised macrocycle are configurationally stable ($\Delta G^{\ddagger} > 80$ kJ mol$^{-1}$). This arises from the increased length of the terphenyl arms, which exceeds the maximum cavity width of the macrocycle (Supp. Figs. 6-2 and 6-3), to prevent intramolecular somersaulting (Supp. Fig. 6-1). This strategy has been exploited for configurationally stable pillar[5]arene derivatives[9,33].

A combination of X-ray crystallography, CD spectroscopy and computational chemistry was required to assign the diastereomers **8a–c**. Macrocycle **8c** readily formed single crystals by vapour diffusion of hexane into a solution of CHCl$_3$ and was therefore identified as the $MP$-isomer by X-ray crystallography (Fig. 2b, Supp. Fig. 5-1, Supp. Data 6)[34]. The crystal structure of **8c** confirmed the PDI's helical core-twist, with a dihedral angle of 20°, in line with other bay-functionalised PDIs[17]. It also revealed a slipped π-stack between the two PDI units, a geometry which facilitates π–π stacking between the naphthalene subunits of PDIs, which have opposite handedness[18]. However, the long-axis displacement of **8c** is small enough (3.7 Å) that any intramolecular excitonic coupling between the chromophores is expected to be H-type[35,36]. $^1H$ NMR spectroscopy also provided evidence for the identity of the $MP$ stereoisomer, as two peaks are observed for $H_a$, corresponding to the $M$ and $P$ helical PDI unit, respectively, which are not equivalent due to the L-valinol imide group. In contrast, these protons give only single peaks in macrocycles **8a** and **8b**, on account of the matching helical chirality of the two PDIs ($MM/PP$).

The chirality of macrocycles **8a** and **8b** was assigned by a combination of CD spectroscopy (vide infra) and computational methods (Supp. Figs. 6-5 to 6-9 and Supp. Data 1 and 2). For the latter, a conformer search using the CREST code[37] and the GFN2-xTB tight binding method[38] was performed for all three stereoisomers of a simplified version of **8** in which the bay substituents were shortened to a single phenyl ring (i.e., macrocycle **5**). To first validate this approach, we optimised the (lowest energy) $MP$-isomer obtained by CREST by density functional theory (DFT-B97-3c[39,40]), which gave a structure in close agreement with that obtained by single-crystal XRD (Supp. Fig. 6-9). Furthermore, time-dependent-DFT (TD-DFT, ωB97x[39,40]) predicted a CD spectrum that best matched the experimental dataset for **8c** (Fig. 3c and Supp Fig. 6-8). From here, we used CREST to predict the lowest energy conformers for both $MM$ and $PP$ isomers, where it was also found that $MM$ lies lower in energy (Supp. Table 6-2). The obtained structure for the $MM$ isomer was subsequently optimised by DFT (B97-3c, Supp. Fig. 6-5) and its spectrum predicted by TD-DFT (ωB97x,

Supp. Table 6-3 and Supp. Fig. 6-6), yielding a negative Cotton-effect and thereby allowing us to assign **8a** as the $MM$ isomer, and the remaining isomer **8b** as $PP$.

The relative yields of the diastereomers of configurationally stable macrocycle **8** are ~ 1:1:2 $MM$:$PP$:$MP$, a statistical outcome that indicates an absence of chiral templation during the (irreversible) macrocyclisation step. However, this does not necessarily imply that these three diastereomers are isoenergetic. Interestingly, analysis of the $^1H$ NMR and CD (vide infra) spectra of dynamic macrocycle **5** (Supplementary Information) revealed an energetic preference for $MP$ over $MM/PP$ stereoisomers (~ 1:0.45, Supp. Fig. 2-5). We attribute this to the relatively bulky aryl-substituents in the PDI's bay positions, which likely sterically hinder co-facial π–π stacking of the PDI units and thus favour slip-stacking, as seen in the X-ray crystal structure of **8c**.

## Photophysical and chiroptical properties

We next investigated the photophysical properties of these macrocycles. From UV-vis spectroscopy (Fig. 3a), a 13 nm redshift of the main PDI absorption band ($S_0 \rightarrow S_1$) is observed upon π-extension of the bay-substituents from a single phenyl ring (PDI monomer **3**) to a terphenyl group (**6**). Both PDI derivatives are monomeric in solution, showing a consistent ratio between 0–0 and 0–1 vibronic peak intensities ($A_{0-0}/A_{0-1}$) across a concentration range of two orders of magnitude (Supp. Figs. 2-1 and 2.2). Upon macrocyclisation, both the phenyl- and terphenyl-systems showed a decrease in the 0-0/0-1 vibronic peak ratio ($\Delta A_{0-0}/A_{0-1} = 0.2$ for **5**, and 0.1 for **8a–c**), consistent with H-type excitonic coupling in the PDI dimers. Intramolecular chromophore coupling is also evidenced by Cotton effects in the CD bands that arise from the $S_0 \rightarrow S_1$ PDI absorption (~500–650 nm). Similar results were found by fluorescence spectroscopy (Fig. 3b), with PDI emission red-shifted both upon macrocyclisation (i.e., from PDI monomer to dimer) and π-extension (from phenyl to terphenyl).

We next investigated the chiroptical properties of macrocycles **5** and **8a–c** and the impact of their various sources of chirality, namely point chirality (L-valinol-based imide groups), intrinsic chromophore helicity (PDI core-twist = $M/P$) and supramolecular chirality (PDI dimer = $M_s/P_s$). Overall, we measured consistently larger $g_{abs}$ values for **8a–c** relative to **5** (Fig. 3c, Table 1, Supp. Figs. 2-3 and 2-4), consistent with the configurational stability of the former. For macrocycle **5**, dynamic chirality leads to the simultaneous population of multiple diastereomeric conformations with opposite CD spectra and hence the cancellation of CD signals. Furthermore, contributions from the intrinsic PDI helicity in **5** are negated due to the $MP$ diastereomer, with its opposing perylene core-twists, being the dominant species in solution.

For this reason, the 'heterochiral' $MP$ diastereomer **8c** serves as a 'chiral control' since its CD spectrum should be dominated by the supramolecular chirality of the PDI dimer, the rotational displacement of which is directed by the point chiral L-valinol-based imide groups[29]. Here, the negative Cotton-effect (for $S_0 \rightarrow S_1$ absorption) indicates the $MP$ PDI dimer has $M_s$ supramolecular chirality in solution[41]. The similarities between the CD spectra (and $|g_{abs}|$ values) of macrocycles **5** and **8c** also evidence the importance of biased supramolecular chirality in defining the chiroptical properties of the heterochiral PDI dimers.

For the 'homochiral' $MM$ and $PP$ diastereomers **8a** and **8b**, the CD spectra have an almost mirror image appearance and yield identical $|g_{abs}|$, evidencing their pseudo-enantiomeric relationship. Notably, the macrocycle's supramolecular chirality is now directed by intrinsic PDI helicity. This is reflected in the appearance of the Cotton effect of **8a,b**. We observe a Cotton-effect for **8a** arising from the rotational displacement of the two dye units (supramolecular chirality), yet there is a clear absence of a Cotton-effect for **8b**. Therefore, while we see matching chirality of $M$ from the PDI and $M_s$ for **8a**, thereby leading to rotational displacement and a Cotton-effect in the CD spectrum of **8a**, we propose that the mismatch of $P$ helicity in the PDI (directing to $P_s$) and the bias towards $M_s$ from the L-valinol groups are competing, leading to effectively co-linear PDIs and an absence

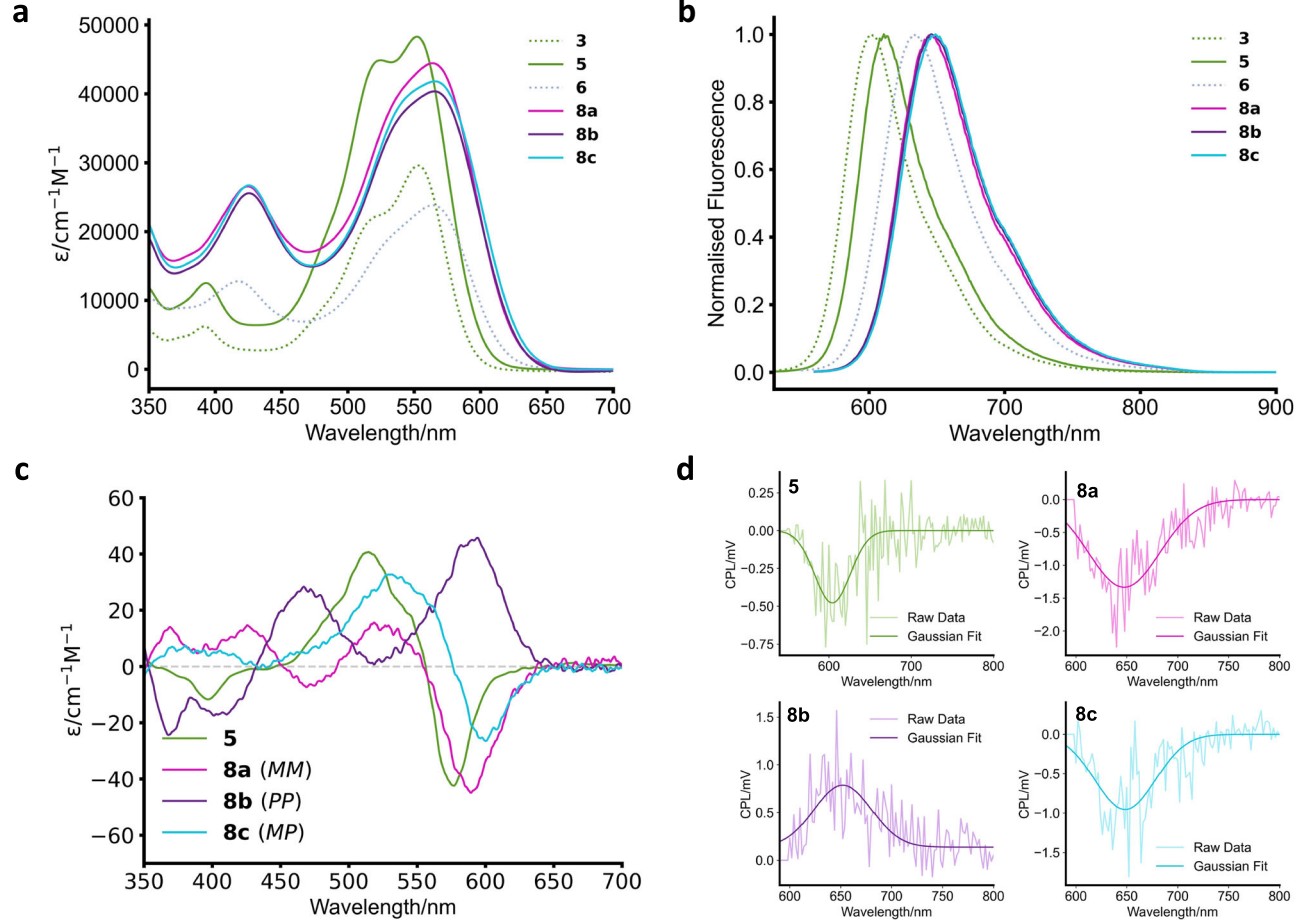

**Fig. 3 | Photophysical and chiroptical properties of the macrocycles. a** Absorption spectra of PDI monomers and macrocycles in CHCl₃ (10 μM). **b** Normalised emission spectra of PDI monomers and macrocycles in CHCl₃ (10 μM). **c** Circular dichroism spectra of macrocycles **5, 8a–c** in CHCl3 (10 μM). **d)** CPL spectra of macrocycles **5, 8a–c** in CHCl₃ (2.5 μM).

**Table 1 | Summary of g-factors for the different macrocycles 5, 8a–c. Errors in $g_{abs}$ and $g_{lum}$ are $\pm 2 \times 10^{-4}$ and $\pm 5 \times 10^{-5}$, respectively**

| Index | Species | PDI helical chirality | $g_{abs} \times 10^{-3}$ | $\lambda (g_{abs})$/nm | $g_{lum} \times 10^{-3}$ | $\lambda (g_{lum})$/nm | Φ | $B_{CPL}$ |
|---|---|---|---|---|---|---|---|---|
| 1 | **5** | ~ 70% MP | −0.8 | 585 | −0.8 | 580 | 1 | 18 |
| 2 | **8a** | MM | −1.5 | 610 | −0.8 | 640 | 0.74 | 12 |
| 3 | **8b** | PP | 1.5 | 610 | 0.5 | 640 | 0.75 | 8 |
| 4 | **8c** | MP | −1.0 | 610 | −0.5 | 640 | 0.84 | 8 |

of a Cotton-effect in the CD spectrum of **8b**. It is also evident that macrocycles **8a,b** possess a larger |$g_{abs}$| than **8c**. Since dimers **8a–c** only exhibit negligible differences in their intramolecular excitonic coupling (vide supra), the existence of PDI intrinsic helical chirality alongside supramolecular chirality in **8a,b** provides a likely explanation for this amplification.

Macrocycles **5** and **8a–c** were also found to exhibit CPL, the signs of which match the lowest energy CD transitions (Fig. 3d, Sup. Figs. 2-6 to 2-9). In line with the findings from CD spectroscopy, |$g_{lum}$| was larger for **8a,b** (MM, PP) relative to **8c** (MP), since the latter has no contribution from the PDI's helical chirality. Between **8a** and **8b**, the MM-isomer (**8a**) was found to have a slightly larger $g_{lum}$, perhaps indicating that the point-chiral imide groups, which, like MM, also favour $M_s$, and so disincentivise a co-linear arrangement known to occur in PDI dimers upon photoexcitation[42]. The |$g_{lum}$| of all macrocycles was found to be smaller than their corresponding |$g_{abs}$|. We attribute this decrease in dissymmetry factor to the absence of bay-strapping in the macrocycles, meaning the PDIs are still able

to core-twist[43], which may lower the contributions of the intrinsic helical PDI chirality.

Interestingly, the $g_{lum}$ for the single phenyl macrocycle **5** was found to be similar to the MM-isomer **8a**, despite the population of competing isomeric forms in **5**. This may be due to a stronger dye-dye interaction compared to the terphenyl system, as evidenced by the larger change in 0-0/0-1 ratio between monomer and macrocyclic dimer.

To further investigate their emissive properties, we performed quantum yield (Φ) measurements on all macrocycles (Table 1 and Supp. Fig. 2-10). We found that between the MM and PP stereoisomers **8a** and **8b**, there was a negligible difference in Φ, whereas the MP stereoisomer **8c** had a higher Φ, likely due to the slipped-stack arrangement of the two PDI units, which reduces H-type coupling to make emission more allowed[11]. Similarly, we found a unity quantum yield for macrocycle **5**, the dominant species of which is the MP stereoisomer. From this data, we also calculated the CPL brightness

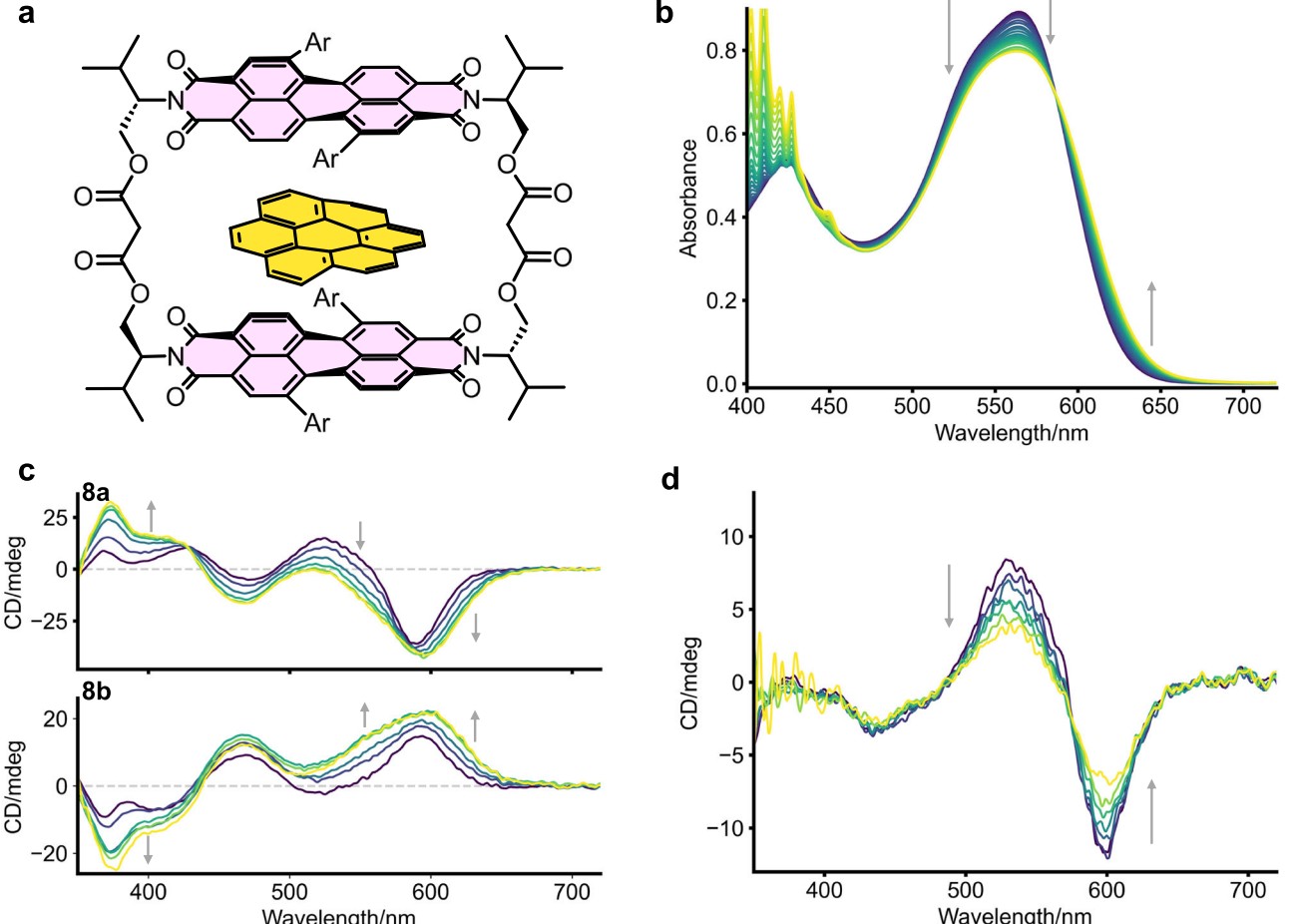

**Fig. 4 | Guest binding inside the core-twisted macrocycles. a** Binding of a planar guest species (coronene) inside a homochiral macrocycle (*MM/PP*), causing chirality transfer to the guest species. **b** Spectral changes in **8a** (10 µM) upon addition of coronene in CHCl₃. **c** Changes in CD of **8a/8b** (CHCl₃, 15 µM) upon addition of coronene. **d** changes in CD of *MP*-isomer **8c** (CHCl₃, 10 µM) upon addition of coronene.

($B_{CPL}$) for the macrocycles (Table 1), which are in line with other organic emitters, including pyrene excimers and helicenes[44].

**Guest binding**

We were curious if these macrocycles, with their core-twisted PDIs, were able to bind polycyclic aromatic hydrocarbon (PAH) guests (Fig. 4a). We initially tested the dynamic, single Ph-substituted macrocycle **5** with several planar PAH guests by UV-vis titrations (Table 2, Fig. 4b, Supp. Figs. 4-1 to 4-3). Indicative of guest binding, a decrease in intensity of the main PDI absorption band, a small decrease in vibronic ratio and the emergence of a charge-transfer (CT) band are observed. The binding strength ($K_a$) of the 1:1 host–guest complex (obtained from non-linear curve fitting using http://app.supramolecular.org/bindfit/ see Supplementary Information section 4) increases with π-extension of the guest from pyrene to perylene to coronene[29]. Interestingly, despite non-planar PDI units, and therefore potentially weaker π-π interactions[13], the macrocycle is still able to bind planar guests, albeit with a lower $K_a$ than a related non-bay-modified bis-PDI macrocycle[29]. We also investigated the binding mode of the strongest bound guest, coronene, by ¹H NMR spectroscopy (Supp. Fig. 7-2), where we saw a clear deshielding of the methylene groups in the malonate linker of **5**, indicating binding of the guest within the macrocycle cavity.

We then tested the strongest bound guest, coronene, with the configurationally stable macrocycles **8a–c** (Supp. Figs. 4-4 to 4-6). We observed slightly stronger binding to the *MM* and *PP* diastereomers **8a** and **8b**, compared to the dynamic macrocycle **5**, but significantly weaker binding to the *MP* diastereomer **8c**. The near-identical binding constant between **8a**

**Table 2 | Binding constants ($K_a$) of macrocycles 5, 8a–c in CHCl₃ with different PAH guests**

| Index | Host | Guest | $K_a$/M⁻¹ | g_lum ×10⁻³ | λ (g_lum)/nm |
|---|---|---|---|---|---|
| 1 | **5** | Pyrene | 40 | - | - |
| 2 | **5** | Perylene | 220 | - | - |
| 3 | **5** | Coronene | 2500 | - | - |
| 4 | **8a** (*MM*) | Coronene | 3700 | −1 | 662 |
| 5 | **8b** (*PP*) | Coronene | 3500 | 1 | 662 |
| 6 | **8c** (*MP*) | Coronene | 600 | −0.2 | 664 |

Values were obtained by non-linear curve fitting of the UV-visible titration data using Bindfit. Fitting errors <1%.

and **8b** is unsurprising, since these *MM* and *PP* macrocycles are pseudo-enantiomeric. The loss of binding affinity for the *MP* diastereomer **8c** likely arises from bringing the PDIs into closer proximity, which would lead to steric repulsion between the (eclipsed) aryl bay-substituents, which is in contrast to **8a/b**, where the PDI aryl groups are staggered, or alternatively is due to the slipped-stack arrangement, which provides a smaller π-surface area for guest binding. Despite the *MP* stereoisomer being dominant in solution for macrocycle **5**, the observation that it possesses a similar $K_a$ as that of the *MM/PP* diastereomers indicates that coronene complexation changes the relative energies of the stereoisomers from *MP* > *MM/PP* to *MM/PP* > *MP*. The fact that these chiral conformers are in thermodynamic

equilibrium in **5**, and their interconversion will come at an energetic cost, explains the weaker binding compared to the configurationally stable, and hence preorganised, diastereomers **8a** and **8b** (*MM* and *PP*).

We next investigated the effect of guest binding on the chiroptical properties of the macrocycles (Fig. 4c, d and Supp. Figs. 4-11 to 4-14). For both **8a** and **8b**, coronene binding causes a loss in the Cotton effect of the main PDI absorption band in the CD spectrum. This suggests an unwinding of the macrocycle dimer (i.e., a reduction in supramolecular chirality)[29] and/ or a reduction in intramolecular electronic coupling upon binding. The monosignate CD ($S_0 \rightarrow S_1$ absorption, ~500–650 nm), therefore, resembles that of a monomeric core-twisted PDI[43], with the +/- sign of the band confirming the chromophore's *P/M* helicity, in line with related PDIs[45]. Notably, however, the $|g_{abs}|$ is similar upon guest binding ($1.7 \times 10^{-3}$ at 610 nm), indicating the importance of the chromophore's intrinsic helical chirality in eliciting a CD spectrum. This is further evidenced by the *MP*-isomer (Fig. 4d), which lacks intrinsic chromophore chirality, and so the full CD spectrum is depleted upon guest binding.

Interestingly, we also observe the emergence of a new CD band at $\lambda$ = ~380 nm (where the PDI only weakly absorbs), the sign of which is opposite for the two diastereomers **8a** and **8b**. We ascribe this new band to bound coronene, since this region of the spectrum is consistent with the lowest energy absorption of the coronene guest, thereby suggesting that chirality is being transferred from the chiral (and configurationally stable) host to the achiral guest upon complexation, as previously reported[21]. This theory is also supported by the observation that chiral imprinting is not observed from the *MP* stereoisomer **8c**, due to the opposite handedness of the two PDI units. The same is also seen for macrocycle **5** (Supp. Fig. 4-14) since in this system the pseudo-enantiomers (*MM* and *PP*) are expected to possess similar $K_a$ values for coronene, and therefore the pseudo-enantiomeric excess is too small to be detected. This result confirms the importance of the intrinsic PDI helical chirality arising from core-twisting to chiral imprinting onto achiral guests.

We finally examined the effect of guest binding on the (circularly polarised) luminescence properties of the macrocycles **5, 8a–c** (Supp Figs. 4-15 to 4-17). In all cases, a decrease in fluorescence emission intensity is observed, as well as a small red shift in $\lambda_{max}$. Despite the reduction of emission, the intensity of the CPL signal is not affected for **8a** and **8b** upon complexation with coronene, whereas for the *MP*-isomer **8c**, the intensity of the CPL signal decreased upon guest binding (as observed also by CD)[29]. Therefore, unlike **8c**, no loss of $g_{lum}$ for **8a** and **8b** was observed upon guest binding (Table 2), again confirming the importance of intrinsic PDI chirality for persistent chiroptical properties.

## Conclusions

We have prepared a new class of bis-PDI-based macrocycles containing multiple sources of chirality (point-chiral linker, helical PDI twist, supramolecular chirality). Small phenyl-substituents in the bay position generated a conformationally dynamic macrocycle, enabling the study of the relative energy of possible diastereomers. Extension of the aryl substituent to terphenyl allowed for the isolation of configurationally stable diastereomers, as it prevents PDI rotation through the cavity. Their diastereotopic relationship, arising from the point chirality in the L-valinol-derived linker, also allowed for their isolation without the need for chiral HPLC.

Solution-phase analysis of the resulting macrocycles showed distinct chiroptical properties arising from the different chiral geometries of the PDI dimers. The configurationally stable macrocycles **8a–c** showed the highest $|g_{abs}|$ as they are diastereomerically pure (compared to dynamic macrocycle **5**). Both the L-valinol's point chirality as well as the PDI's intrinsic helical chirality impact the supramolecular chirality, with the former found to be a dominant factor. The *MM/PP* isomers have identical $|g_{abs}|$ values, which were higher than the *MP*-isomer **8c**, which overall lacks intrinsic PDI chirality.

The macrocycles were also able to bind (achiral) PAH guests. The homochiral macrocycles *MM/PP* showed stronger binding than the heterochiral macrocycle, indicating the importance of staggered over eclipsing bay substituents. The homochiral *MM/PP* macrocycles maintained their CD/CPL signal upon guest binding and enabled chiral imprinting onto the guest. This behaviour was not observed with heterochiral macrocycle **8c** or dynamic macrocycle **5**, where only a loss of signal was observed. This demonstrates the importance of stable and intrinsically chiral chromophores for maintaining and imparting chiroptical properties upon combination with achiral guest species.

In summary, these results reveal the impact of multiple sources of chirality on molecular conformation, guest binding and chiroptical properties and will inform the future development of chiroptical materials.

## Supplementary information

Synthetic procedures for compounds **1**–**8**, characterisation data, additional spectra, titration data, single-crystal diffraction data, DFT optimised structures, and experimental and computational methods can be found in the Supplementary Information. Deposition Number 2489527 contains the supplementary crystallographic data for this paper. These data are provided free of charge by the joint Cambridge Crystallographic Data Centre and Fachinformationszentrum Karlsruhe Access Structures service.

## Methods

All synthetic methods and characterisation for compounds **1-8**, spectroscopic, titration, crystallography and computational methods can be found in the supplementary information.

## Data availability

The data that support the findings of this study are available in the supplementary material of this article. Raw data can be obtained from the authors upon reasonable request. The supplementary data files include the DFT optimised structures of macrocycle **5** *MM* stereoisomer (Supplementary Data 1) and *MP* stereoisomer (Supplementary Data 2), a Ph-substituted PDI monomer *M* stereoisomer (Supplementary Data 3) and *P* stereoisomer (Supplementary Data 4), and the transition state for the interconversion between these *P* and *M* PDI monomer stereoisomers (Supplementary Data 5). The X-ray crystallographic coordinates for structures reported in this Article have been deposited at the Cambridge Crystallographic Data Centre (CCDC), under deposition number CCDC 2489527. These data can be obtained free of charge from The Cambridge Crystallographic Data Centre via www.ccdc.cam.ac.uk/data_request/cif. The cif for macrocycle **8c** is included in the supplementary data files (Supplementary Data 6).

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

## Acknowledgements

T.A.B. and D.H. thank the EPSRC (EP/W037661/1) for funding. R.P. and S.P. thank the BBSRC (BB/S017615/1 and BB/X001172/1) and the EPSRC (EP/X040259/1) for funding. T.A.B. and D.H. thank the EPSRC National Crystallography Service for collecting single-crystal X-ray diffraction data.

## Author contributions

D.H. and T.A.B. devised the project. D.H. performed the majority of the work. S.E.P. and R.P. performed the CPL measurements. D.H. and T.A.B. wrote the manuscript.

## Competing interests

The authors declare no competing interests.
