## [Transparent Peer Review file · Communications Chemistry]

Chirally locked and dynamic bis-perylene diimide macrocycles with multiple sources of chirality

Corresponding Author: Dr Timothy Barendt

Version 0:

Reviewer comments:

Reviewer #1

(Remarks to the Author)

This manuscript addresses a timely and important topic, tackling a central challenge in supramolecular chemistry and chiral materials—namely, how to deconvolute and understand the contributions of different sources of chirality (point, helical, and supramolecular) at both molecular and supramolecular levels.

1) In the discussion of supramolecular chirality, compounds 8a and 8b correspond to PP or MM interactions, while 8c involves PM interactions. These distinctions are expected to result in significant structural and chiroptical differences. I strongly recommend that the authors attempt to separate the P and M enantiomers of compound 6 and investigate their interactions individually. Such an approach would be highly valuable for achieving a more quantitative understanding of the interrelationships among different chirality origins.

2) Upon guest binding, the supramolecular chirality would typically be expected to be quenched. However, significant supramolecular chirality is still observed for 8a, 8b, and 8c, which seems disproportionate to the expected quenching effect. The authors are encouraged to provide an explanation or possible rationale for this observation.

3) The authors are encouraged to ensure that the terminology related to chirality is used rigorously and consistently throughout the manuscript. For instance, helical chirality should refer specifically to the molecular chirality arising from the twisted PDI units themselves, whereas supramolecular chirality should denote the chirality expressed at the level of macrocyclic dimers or higher-order aggregates.

Reviewer #2

(Remarks to the Author)

In this manuscript, Hartmann, Penty, Pal and Barendt report on a chiral PBI based macrocycle. The PBIs in the macrocycle display helical chirality while the linkers connecting the macrocycle display point chirality. This study investigates the impact of these two sources of chirality on the overall chirality exhibited by the macrocycle. The authors show that the dominant effect on the chiroptical properties arise from the helical chirality of the PBI molecules.

They show that moving from a phenyl-substituent in the bay-position to a terphenyl-substituent renders the helical chirality of the PBI configurationally stable. This is quite an interesting point as it is not immediately clear how extension in this dimension away from the PBI would render it more configurationally stable. As the main crux of the paper involves this configurational locking, additional explanation about this aspect should be presented (perhaps computationally).

The authors also look at the chiroptical properties. They find that the systems emit CPL, which is not surprising, and the g-factors are on the order of normal small chiral molecules. The changes in chiroptical properties between the samples are used to explain differences in molecular conformations and appear reasonable. Guest binding is then explored for planar aromatic guests. They observed differences in binding constant for the MM (PP) macrocycle compared to the MP diastereomer. Again, they see that there is some transfer of the chiral environment from the macrocycles cavity to the guest, and is not observed for PM macrocycle due to competing handedness.

Overall, this work will be of interest to those working on PBI dyes and chiral materials. I think the manuscript may be suitable for publication if the comments above and below are addressed (providing additional conceptual novelty compared to that already published).

Comments:

1. The authors should take more care in the wording and claims made. “two chiral, core-twisted PDIs connected solely via the imide positions; the first of its kind...”, but the previous work by Würthner (cited as 26) also shows this, connected by

cyclisation at the bay positions, as the authors state, but this is still meeting the requirement of “core-twisted PDIs connected solely via the imide positions”.

2. Figure 1 shows “Larger g-factors, improved Ka, chiral imprinting”, this figure might be revised to show that this is referring to mismatched PDI chirality and matched PDI chirality as it is not evident at the moment what the comparison is (only when you carefully read the caption).

3. Figure 2b, it is hard to read the distance shown in the x-ray.

4. The extinction coefficient of the PBIs of the monomers compared to the macrocycles are far from being 2x the strength. The monomers themselves also appear to aggregate in solution (not very distinct vibronic transitions)? The authors should provide some additional information as to the state of the monomers at this concentration, and the differences in the extinction coefficient.

5. It's not clear from figure 1 what the difference between 8a,b,c is.

6. Are the authors able to use the cavity for deracemization of guest molecules? Similar to the behaviour shown by Würthner in ref 26? This is then more of a useful application.

Reviewer #3

(Remarks to the Author)

In this manuscript, T. A. Barendt and their colleagues designed a series of bis-perylene diimide-based macrocycles containing multiple sources of chirality, specifically point chirality in the linker, helical chirality in the perylene diimide and supramolecular chirality in the macrocyclic dimer. The authors systematically tested the CD and CPL spectra of these compounds, and the data showed that the different chiral geometries of PDI dimers produced different chiral optical properties. In addition, the host-guest recognition of these macrocyclic compounds with (achiral) PAHs has also been studied. The inherent chiral chromophores of macrocycles can impart chiral properties to guest molecules when combined with achiral PAHs. Overall, these results reveal the influence of multiple chiral sources on conformation, chiral properties, and guest binding. This paper can be considered for publication, but some necessary problems should be addressed. I suggest revision and re-review this manuscript.

1. In the references, you cited relevant papers on chiral compounds with CPL activity. However, there is a missing citation of the pioneering work of Chuan-Feng Chen, Pangkuan Chen et al. on chiral macrocyclic arenes (Angew. Chem. Int. Ed. 2021, 60, 21927-21933; Angew. Chem. Int. Ed. 2020, 59, 11267-11272; Chem. Sci., 2023,14, 987-993), which involves the influence of guest molecules on the chiroptical properties of macrocyclic arenes.

2. The author mentioned that compound 8 has better conformational stability compared to compound 5, but did not provide data on the energy barriers for conformational interconversion of these compounds. Perhaps DFT calculations can be used to obtain these data and further compare the stability of their configurations.

3. What does ΔG^\ddagger represent? Which configuration does it represent the free energy of transition to which configuration?

4. In Figure 2a, the structures of the compounds are not clear enough.

5. How is the binding ratio of host and guest (1:1) determined? Please provide corresponding experimental data.

6. Fluorescence quantum yield (Φ) and CPL brightness (BCPL: defined as $BCPL = \epsilon \times \Phi \times |g_{lum}|/2$) is an important parameter for CPL-active materials. Please provide the corresponding data in Table 1.

7. The concentration can affect the intermolecular stacking, which in turn affects the chiral optical signal. Please provide the concentration for spectral testing in Figure 4.

8. Please provide the g_{lum} factor of the host-guest complex in Table 2.

9. The change in chemical shift of host-guest NMR can be used to infer the binding between the guest molecule and the macrocycle. Please provide the host-guest 1H NMR spectrum.

Version 1:

Reviewer comments:

Reviewer #1

(Remarks to the Author)

I am satisfied with the authors' revisions, which have fully addressed the referee's previous comments. I recommend that the manuscript be accepted for publication.

Reviewer #2

(Remarks to the Author)

The authors have addressed my comments. I therefore recommend publication of this work.

Reviewer #3

(Remarks to the Author)

After the authors' careful revisions, the quality of the manuscript has been significantly improved, and I agree to its publication.

Our responses to the comments from each of the reviewers in turn:

Reviewer #1:

1. *In the discussion of supramolecular chirality, compounds 8a and 8b correspond to PP or MM interactions, while 8c involves PM interactions. These distinctions are expected to result in significant structural and chiroptical differences. I strongly recommend that the authors attempt to separate the P and M enantiomers of compound 6 and investigate their interactions individually. Such an approach would be highly valuable for achieving a more quantitative understanding of the interrelationships among different chirality origins.*

We thank the reviewer for this suggest, however we are unable to separate the *M* and *P* (dia)stereoisomers of **6** due to their low racemisation barrier, which is a common trait for acyclic core-twisted PDIs (see *J. Am. Chem. Soc.* **2007**, 129, 14319), hence the requirement for the configurationally stable macrocycle **8** developed here. To support this assessment, we have now calculated the interconversion barrier for a phenyl-substituted isomer of acyclic PDI **6** and found it to be 34 kJ/mol (new Supplementary Figure 6-4 and revisions on page 3); too low for enantiomer resolution according to the reference above. Furthermore, to provide clarification for how macrocycle **8** provides configurational stability, we have included additional discussion in the manuscript (page 4) and a new figure in the SI (Supplementary Figure 6-1), to explain the mechanism for stereoisomer interconversion in the bis-PDI macrocycles.

2. *Upon guest binding, the supramolecular chirality would typically be expected to be quenched. However, significant supramolecular chirality is still observed for 8a, 8b, and 8c, which seems disproportionate to the expected quenching effect. The authors are encouraged to provide an explanation or possible rationale for this observation.*

The reviewer is correct in their assessment and indeed a loss of supramolecular chirality is observed in **8a-c** upon binding, as evidenced by the depletion of the Cotton effect in their CD spectra (Figure 4c,d). This is most clearly seen in **8c** (Figure 4d), the macrocycle where only supramolecular chirality is responsible for the CD spectrum, and hence the entire CD spectrum decreases upon complexation as anticipated. For **8a** and **8b**, while we observe a loss of supramolecular chirality (again indicated via the loss of the Cotton effect, Figure 4c), we retain the helical chirality of the twisted PDI units (*M* or *P*); therefore we are still observing a CD spectrum, which corresponds to monomeric/molecular PDI chromophores. This explanation is provided on page 10 (paragraph 1) in the manuscript, but we have made a minor revision to clarify that the full CD spectrum of macrocycle **8c** (*MP*) is depleted, due to it only possessing supramolecular chirality.

3. *The authors are encouraged to ensure that the terminology related to chirality is used rigorously and consistently throughout the manuscript. For instance, helical chirality should refer specifically to the molecular chirality arising from the twisted PDI units themselves, whereas supramolecular chirality should denote the chirality expressed at the level of macrocyclic dimers or higher-order aggregates.*

We thank the reviewer for bringing this to our attention and we have now revised terminology relating to chirality across the manuscript in line with the definitions above. This includes the use of the term “rotational displacement” (instead of “helicity”) to describe the arrangement of the PDI units that gives rise to the supramolecular chirality in the macrocycle dimers.

Reviewer #2:

1.1. *They show that moving from a phenyl-substituent in the bay-position to a terphenyl-substituent renders the helical chirality of the PBI configurationally stable. This is quite an interesting point as it is not immediately clear how extension in this dimension away from the PBI would render it more configurationally stable. As the main crux of the paper involves this configurational locking, additional explanation about this aspect should be presented (perhaps computationally).*

We thank the reviewer for raising this point and so, to provide additional clarity, we have now included new diagrams that explain the mechanism of interconversion between macrocycle stereoisomers (a revised Figure 1 and new Supplementary Figures 6-1, 6-2 and 6-3). This is underpinned by a new discussion in the manuscript as to how bay extension affords configurational stability (page 4, paragraph 2). We have also performed new calculations on the interconversion barrier between *M* and *P* PDI stereoisomers, which support our experimental results (see Supplementary Figure 6-4 and a new discussion on page 3, paragraph 5). In summary, the interconversion barrier between *M* and *P* stereoisomers is relatively small (34 kJ/mol). However, this motion alone does not lead to enantiomer interconversion in the macrocycle, since it must be coupled to a somersault of the PDI through the cavity of the macrocycle, which is prevented by elongation of the bay substituents. Therefore, the barrier to somersaulting is the energetic barrier measured experimentally by ¹H NMR spectroscopy. This point has been clarified by revisions to paragraph 1 on page 4.

1.2 *The authors should take more care in the wording and claims made. “two chiral, core-twisted PDIs connected solely via the imide positions; the first of its kind...”, but the previous work by Würthner (cited as 26) also shows this, connected by cyclisation at the bay positions, as the authors state, but this is still meeting the requirement of “core-twisted PDIs connected solely via the imide positions”.*

We thank the reviewer for their guidance here. Indeed, we have corrected this sentence (page 2, paragraph 3) to distinguish our macrocycle design from that in reference 26: “In this work, we aimed to prepare the first configurationally stable macrocycles composed of two chiral, core disubstituted PDIs connected via the imide positions and without additional covalent strapping over the PDI units, as has previously been employed in bis-PDI macrocycles.^{21,28}”

2. *Figure 1 shows “Larger g-factors, improved Ka, chiral imprinting”, this figure might be revised to show that this is referring to mismatched PDI chirality and matched PDI chirality as it is not evident at the moment what the comparison is (only when you carefully read the caption).*

We thank the reviewer for this suggestion and have revised Figure 1 for clarity.

3. *Figure 2b, it is hard to read the distance shown in the x-ray.*

We have amended Figure 2b to make this distance easily readable, thank you.

4. *The extinction coefficient of the PBIs of the monomers compared to the macrocycles are far from being 2x the strength. The monomers themselves also appear to aggregate in solution (not very distinct vibronic transitions)? The authors should provide some additional information as to the state of the monomers at this concentration, and the differences in the extinction coefficient.*

We thank the reviewer for pointing this out. We have now measured the absorption spectra for both PDI derivatives at three different concentrations, covering two orders of magnitude (see new Supp. Figures 2-1 and 2-2). These compounds show no change in the 0-0/0-1 ratio

of the main PDI absorption band, which evidences an absence of intermolecular aggregation (clarified by a revision to paragraph 4, page 5). These measurements enabled us to obtain accurate extinction co-efficients (ϵ) and Figure 3a has been revised accordingly. Indeed, the macrocycles have ϵ values that are close to double those of their respective monomers (e.g., $\epsilon = 24,000$ vs $42,000$ at 564 nm for **6** and **8a-c**, respectively). The observation that macrocycle ϵ is slightly less than two times that of the monomer has previously been observed in bis-PDI macrocycles by us (*Angew. Chem. Int. Ed.* **2025**, e20250112) and others (*Angew. Chem. Int. Ed.* **2015**, 10165, see Supp. Figures S3 and S4), and is likely due to intramolecular electronic coupling between the two PDI chromophores. Furthermore, the less well-defined vibronic transitions observed are also common for bay-arylated PDIs as reported in *J. Org. Chem.* **2005**, 4323.

5. It's not clear from figure 1 what the difference between **8a,b,c** is.

We thank the reviewer for bringing this to our attention. We have amended Figure 1 with new labels to identify macrocycles **8a-c**. We have also added the PDI chirality labels to Figures 2a and Figure 3c for clarity.

6. Are the authors able to use the cavity for deracemization of guest molecules? Similar to the behaviour shown by Würthner in ref 26? This is then more of a useful application.

We thank the reviewer for this suggestion and so we tested the deracemization of [5]-helicene with **8a** (*MM*), following similar experimental conditions stated in the original ref. 26 (see below). We measured the CD of **8a** in CHCl_3 at $100 \mu\text{M}$ before addition of [5]helicene (black arrow) and then monitoring over 24 hours, where we unfortunately saw no evidence of deracemization, as the CD signal at 310 nm remained unchanged. Our current hypothesis is that this is due to weaker binding since the cavity of macrocycle **8** is larger and more flexible than that in ref. 26. While we believe that further work is beyond the scope of this manuscript, we are currently investigating the enantioselective synthesis of chiral polycyclic aromatic hydrocarbons from achiral precursors using a different bis-PDI macrocycle design, which will be the subject of a new paper. The work in this paper remains valuable to the supramolecular and chiral materials communities because we establish the impact of point/helical/supramolecular chirality on fundamental structure-property relationships.

Reviewer #3:

1. *In the references, you cited relevant papers on chiral compounds with CPL activity. However, there is a missing citation of the pioneering work of Chuan-Feng Chen, Pangkuan Chen et al. on chiral macrocyclic arenes (Angew. Chem. Int. Ed. 2021, 60, 21927-21933; Angew. Chem. Int. Ed. 2020, 59, 11267-11272; Chem. Sci., 2023, 14, 987-993), which involves the influence of guest molecules on the chiroptical properties of macrocyclic arenes.*

We thank the reviewer for suggesting these relevant references and we have included them in our revised introduction and in our discussion on the chiral stability of macrocycles **8a-c** (new references 8,9 and 33).

2. *The author mentioned that compound 8 has better conformational stability compared to compound 5, but did not provide data on the energy barriers for conformational interconversion of these compounds. Perhaps DFT calculations can be used to obtain these data and further compare the stability of their configurations.*

Our understanding is that this issue is also related to the reviewer's 3rd point and indeed those raised by other reviewers (#1 point 1, #2 point 1.1). Therefore, we performed new DFT calculations to show that interconversion of the PDI's helical chirality (*M/P*) is dynamic (Supplementary Figure 6-4). This means that it is the intramolecular somersault of the PDI through the cavity of the macrocycle that governs the energetic barrier to stereoisomer interconversion, as outlined in revisions to the manuscript (paragraph 5, page 3 and paragraph 2, page 4). The energetic barriers to PDI somersaulting had been calculated by variable temperature ¹H NMR spectroscopy ($\Delta G^\ddagger = 78 \text{ kJ mol}^{-1}$ for **5**, and $> 80 \text{ kJ mol}^{-1}$ for **8**), which evidences that elongation of the bay substituents in macrocycle **8** is responsible for configurational stability. We have included new Figures (Supplementary Figures 6-1 to 6-3) and a revision to Figure 1, to explain how this elongation impacts the mechanism for stereoisomer interconversion. Additional text in the manuscript (page 4, paragraph 1, and page 4, paragraph 2) will also make these points clearer for the reader.

3. *What does ΔG^\ddagger represent? Which configuration does it represent the free energy of transition to which configuration?*

We hope that our response to the above point addresses these questions and that the new Figures and text in the manuscript provide improved clarity for the reader. We have included the ΔG^\ddagger labels on Figure 1 to make it clearer that this barrier represents the barrier to interconversion between macrocycle stereoisomers (*PP/PM/MM*), which is governed by the intramolecular somersaulting motion in the mechanism (new Supp. Fig. 6-1) and now supported computationally (new Supp. Fig. 6-3).

4. *In Figure 2a, the structures of the compounds are not clear enough.*

This point was also raised by reviewer #2 (point 5) and we have now ensured that the chirality of macrocycles **8a-c** are appropriately labelled in revisions to both Figure 2a and Figure 1. We also increased the size of the synthetic scheme in Figure 2a and added new chirality labels in Figure 3c for clarity.

5. *How is the binding ratio of host and guest (1:1) determined? Please provide corresponding experimental data.*

To ensure data fitting to the correct binding stoichiometry, we explored 1:1, 1:2 and 2:1 host-guest models and found the best fit is for the 1:1 model in all cases. As an example, the binding of macrocycle **5** with coronene only gave sensible data with a 1:1 model (please see the following link to the fit: <http://app.supramolecular.org/bindfit/view/6019ca27-3d54-4917-95ca->

[47431c5e8169](http://app.supramolecular.org/bindfit/view/8580a992-ddfa-4251-98d6-0e4d78878aab)). A 1:2 model yielded a nonsensical negative K_{12} value (<http://app.supramolecular.org/bindfit/view/8580a992-ddfa-4251-98d6-0e4d78878aab>), and a 2:1 model yielded higher errors and an excessively large K_{12} (<http://app.supramolecular.org/bindfit/view/013581f7-c039-45f7-a242-046866ebac21>). These methods for data fitting have been clarified to the reader by new text at the start of Section 4 in the Supporting Information. All titrations are accompanied by a hyperlink (as above) to the fit obtained via the Bindfit programme at supramolecular.org/bindfit/, which is recommended by the programme developer as the best way to provide open access binding data (*Chem. Commun.*, **2016**, 12792). This link provides information on the model used, the fit to the data and the corresponding error and residuals.

6. *Fluorescence quantum yield (Φ) and CPL brightness (BCPL: defined as $BCPL = \epsilon \times \Phi \times [glum]/2$) is an important parameter for CPL-active materials. Please provide the corresponding data in Table 1.*

We agree with the reviewer that B_{CPL} is an important measure of CPL activity and have therefore measured quantum yields (Φ) and corresponding B_{CPL} values, all which have been added to Table 1.

7. *The concentration can affect the intermolecular stacking, which in turn affects the chiral optical signal. Please provide the concentration for spectral testing in Figure 4.*

We have added the concentrations used in the measurements to the captions for Figures 3 and 4, thank you.

8. *Please provide the $glum$ factor of the host-guest complex in Table 2.*

We have added the g_{lum} values to Table 2 as requested and revised the text to direct the reader to them (paragraph 3, page 10).

9. *The change in chemical shift of host-guest NMR can be used to infer the binding between the guest molecule and the macrocycle. Please provide the host-guest 1H NMR spectrum.*

We thank the reviewer for this suggestion and agree with them as to the importance of investigating the binding mode via 1H NMR spectroscopy. We have therefore titrated macrocycle **5** with 0.5 and 1 equivalents of coronene (1.6 mM concentration in $CDCl_3$) and saw a clear shift of the protons corresponding to the malonate linker methylene group, thereby indicating binding within the macrocycle cavity. We have included this data in the SI as Supplementary Figure 7-2 and added a new discussion of this result in the manuscript (paragraph 2, page 8).